

# Recovery of antimicrobial susceptibility in methicillin-resistant *Staphylococcus aureus* (MRSA): a retrospective, epidemiological analysis in a secondary care hospital, Sapporo, Japan

Yuji Koike[1,2] and Hiroshi Nishiura[1,3]

[1] Graduate School of Medicine, Hokkaido University, Sapporo, Hokkaido, Japan
[2] Department of Microbiology, JR Sapporo Hospital, Sapporo, Hokkaido, Japan
[3] School of Public Health, Kyoto University, Kyoto, Kyoto, Japan

## ABSTRACT

Anti-methicillin-resistant *Staphylococcus aureus* (MRSA) drugs are critical final options for treating MRSA infection. This study investigated the percentage of all *S. aureus* isolates that are resistant to methicillin and also MRSA susceptibility to other antimicrobial agents in the JR Sapporo Hospital inpatient service. The inpatient service MRSA percentages for Japan, Hokkaido, and JR Sapporo Hospital from 2010–2019 were compared, exploring the annual rate of change in the MRSA percentage. We also investigated the antimicrobial use density (AUD) and its relationship with MRSA antimicrobial susceptibility in the JR Sapporo Hospital during 2019. The MRSA percentage in JR Sapporo Hospital was 61.5% (95% CI [52.6–69.7]) in 2010 but was only 51.6% (95% CI [41.6–61.5]) in 2019, which is a 1.43% (95% CI [0.42–2.43]) annual decrease ($p = 0.05$). Regarding the MRSA antimicrobial susceptibility rate in JR Sapporo Hospital, the highest rates of annual increase were seen for minocycline (3.11% (95% CI [2.25–3.94])) followed by fosfomycin (2.85% (95% CI [1.83–3.85])). Positive correlations with the AUD of anti-MRSA drugs were identified for susceptibility to erythromycin ($p < 0.01$), clindamycin ($p = 0.002$), and levofloxacin ($p = 0.0005$). A recovery of MRSA antimicrobial susceptibility was observed in our antibiogram dataset. Our study supports the potential for appropriate antimicrobial agent use in reviving MRSA antimicrobial susceptibility.

## INTRODUCTION

*Staphylococcus aureus*, a Gram-positive coccus, is harboured by 20–40% of the population, usually acting as normal flora, but also frequently infecting skin, wound, blood stream, and catheters from invasive treatment (e.g. central vein). In particular, methicillin-resistant *Staphylococcus aureus* (MRSA), which possesses a *mecA* gene associated with the production of Penicillin Binding Protein 2 prime (PBP2′), is globally recognized as the most widespread antimicrobial resistance problem, especially in healthcare facilities

Corresponding author
Hiroshi Nishiura,
nishiurah@gmail.com

(*Deresinski, 2005*; *Kale & Dhawan, 2016*; *Lee et al., 2018*). There has been growing concern over the spread of resistance against anti-MRSA drugs in *S. aureus*, and the frequency of fatal outcomes in MRSA-infected cases has been estimated as 64% greater than that in methicillin-sensitive *S. aureus* (MSSA)-infected cases (*World Health Organization, 2020*). MRSA is epidemiologically widespread, not only in Western countries but also in Asia (*Chen & Huang, 2014*). In July 2007, the Ministry of Health, Labour and Welfare of Japan first devised a nosocomial infection surveillance system, known as the Japan Nosocomial Infections Surveillance (JANIS), which provides up-to-date information regarding the epidemiological situation of MRSA (*Ministry of Health, Labour and Welfare, 2020a*). As of October 2020, vancomycin-resistant *S. aureus* (VRSA) has yet to be identified in Japan (*IDSC, 2020*).

MRSA is the primary cause of nosocomial infection; because the risk of death in cases of blood stream infection with MRSA is known to be greater than that for similar cases of MSSA, treatment is required immediately upon diagnosis (*Tan et al., 2001*; *Hassoun, Linden & Friedman, 2017*). The successful treatment of *S. aureus* is highly dependent on the choice of antimicrobial agents (*van Hal et al., 2012*), but the antimicrobial agents that can be used is limited. Recommendations regarding the appropriate use of antimicrobial agents are available (*Ministry of Health, Labour and Welfare, 2016*; *Abubakar & Sulaiman, 2018*), and anti-MRSA drugs are regarded as critical final options for treating MRSA infection (*Chen et al., 2019*). While an increase in antimicrobial-resistant strains of Gram-negative rods (e.g. fluoroquinolone resistance in *Escherichia coli*; *Terahara & Nishiura, 2019*) has emerged as a significant health issue in Japan, MRSA is another pressing problem with a high prevalence. Japan formulated and announced the 2020 antimicrobial resistance (AMR) action plans in recognition of the 2016 Global Action Plan that was approved by the World Health Assembly in May 2015. Accordingly, appropriate antimicrobial agent use was promoted, aiming to reduce the percentage of *S. aureus* that are MRSA from 48.5% in 2015 to less than 20% by 2020 (*Ministry of Health, Labour and Welfare, 2016*).

In Japan, when a causative agent has yet to be identified during the treatment of a bacterial infection, the use of an antibiogram from the applicable regional healthcare facility is recommended to design a suitable empiric therapy (*Niki et al., 2017*). Despite the frequent use of antibiograms for determining treatment options, the associated dataset has yet to be widely used for epidemiological studies (*Kim, Yoo & Chang, 2020*). At JR Sapporo Hospital, a private secondary care hospital in the capital city Sapporo of the northernmost prefecture, Japan, the infection control team was founded in 1987, and since 2005, it has involved physicians, nurses, pharmacists, and clinical laboratory technologists. There have been very few infection control doctors (ICD) who truly act as infection disease specialists in Japan (*Kishida & Nishiura, 2020*), and the ICDs of JR Sapporo Hospital have been concurrently covered by expert physicians in other specialty areas (e.g. a cardiologist and renal physician, as of October 2020). To address its limitation in specialty knowledge, the ICT has involved an infection control nurse since 2014, an infection control microbiological technologist (ICMT) since 2017, and an infectious disease chemotherapy pharmacist since 2018. The team has continuously assessed the antimicrobial susceptibility of bloodstream infections and drug-resistant bacterial agents

upon identification, providing regular surveillance information to each member of the hospital team. In addition, broad-spectrum antibacterial agents and anti-MRSA drugs are managed on a notification basis. The impact of the ICT activities on promoting appropriate antimicrobial agent use for inpatient treatment has yet to be explicitly assessed.

The present study aimed to investigate the percentage of *S. aureus* isolates caused by MRSA, the antimicrobial susceptibility of MRSA isolates, and the relationship between AUD and the antimicrobial susceptibility of MRSA in the JR Sapporo Hospital inpatient service. Here, we comprehensively analyse the dataset from 2010–2019.

## MATERIALS & METHODS

### Study setting

Hokkaido is the second largest island of Japan, located in the northernmost part, and Sapporo city is its capital city, populated by approximately 1.9 million people. JR Sapporo Hospital is within a walking distance from the main station, Sapporo station, and the total inpatient capacity for providing secondary acute care is 312 beds. Additionally, the hospital is designated as the Hokkaido Affiliated Hospital for Cancer Treatment; it also contributes to the primary healthcare service for the region, reserving an independent ward for primary healthcare.

### Data source at JR Sapporo hospital

We aimed to analyse the percentage of *S. aureus* involving MRSA, the time-dependent changes in the percentage of MRSA resistant to specific antimicrobial agents, and the time-dependent changes in the use frequency of various antimicrobial agents. From 2010–2019, JR Sapporo Hospital collected antimicrobial susceptibility data, which was summarized as the Weekly Report for internal use (e.g. during ICT meetings or printed in the Weekly Laboratory Surveillance). The present study used these published secondary data.

Antibiogram data, classified by antimicrobial agent and patient service (i.e. inpatient or outpatient), were collected. As control data, we also extracted the openly accessible data from JANIS (*Ministry of Health, Labour and Welfare, 2020a*), which has conducted a nationwide laboratory-based surveillance of antimicrobial resistance since 2000. As of 2019, a total of 2,100 healthcare facilities with 200 or more beds have been registered with JANIS (*Ministry of Health, Labour and Welfare, 2020b*). Regarding the JANIS data, we retrieved data from not only the entire country but also from just the Hokkaido region to use for comparisons. For both the JR Sapporo Hospital and JANIS datasets, the antimicrobial susceptibility records did not specify whether the patient was a carrier or was actively infected; additionally, no record of sampling part of the body (i.e. infection location) was available. Here, we focused on inpatient data because the impact of ICT activities should be better reflected by the inpatient data than by outpatient data. The applied definition of MRSA adhered to the standardized criteria published by the Clinical & Laboratory Standards Institute (CLSI), M100-S22. In JR Sapporo hospital, we have consistently defined MRSA as resistant to oxacillin (MPIPC) $\geq 4\,\mu g/mL$ or Cefoxitin

(CFX) $\geq$ 8 μg/mL during minimal inhibitory concentration (MIC) susceptibility testing throughout the entire period of our study.

AUD was expressed as defined as the daily dose (DDD) per 1,000 patient-days. The basic definition of DDD is the assumed average maintenance dose per day for a drug used for its main indication in adults. Consequently, the AUD was calculated as (total antimicrobial dose)/(DDD × monthly number of inpatients) × 1,000 (*Lee et al., 2007*; *Yoshida et al., 2013*). The annual number of inpatients was recorded by the accounting section of the hospital.

## Statistical analysis

Three different analyses were conducted. First, we examined the time-dependent changes in the antimicrobial resistance rate of *S. aureus* over the 10-year period from 2010–2019. We measured the resistance rate as the percentage of all *S. aureus* isolates that are resistant to methicillin. To detect trends in the percentage of antimicrobial-resistant *S. aureus*, we devised two simple models, i.e. models with or without a time-dependent linear component, and then employed the likelihood ratio test so that the presence of a trend can be identified by the significant improvement in the likelihood following the incorporation of the trend into the model.

Second, the antimicrobial susceptibility rate of MRSA over time was examined; this was similarly measured as the percentage of MRSA that were also resistant to other antimicrobial agents. For calculating the 95% confidence interval (CI) of the percentage, we employed the Agresti's score confidence interval because the sample sizes were not necessarily large and the susceptibility rates frequently had values close to 0% or 100%. The presence of a time-dependent trend was also judged by a likelihood ratio test.

Third, we examined the relationship between the AUD and the MRSA antimicrobial susceptibility rate by clinical department, so that we could investigate if the use of anti-MRSA drugs is associated with the antimicrobial susceptibility rate. The relationship between these two continuous variables was examined by linear correlation testing, and *p*-values of less than 0.05 were regarded as significant. All statistical analyses were conducted by using JMP ver 14.0.0 (SAS Institute Inc., Cary, NC, USA).

## Ethical considerations

The present study was based on a secondary dataset in which all patient information was deidentified. Therefore, the present study did not require informed consent. This study was approved by the institutional review board of the Hokkaido University Graduate School of Medicine (Med 20-021).

## Data sharing

In the present study, we retrospectively analysed two different datasets. The dataset from the JANIS database can be retrieved online (*Ministry of Health, Labour and Welfare, 2020a*; https://janis.mhlw.go.jp/report/kensa_prefectures.html). The other dataset containing the AUD by department and antimicrobial susceptibility in methicillin-resistant

*Staphylococcus aureus* (MRSA) by department in *JR Sapporo Hospital (2019)* are uploaded as Tables S1 and S2.

## RESULTS

Table 1 shows the annual trends in the number of MRSA isolates and their proportion among *S. aureus* isolates across Japan, for the Hokkaido region, and in JR Sapporo Hospital, Sapporo, Japan. In 2019, 48.1% (95% CI [47.9–48.3]) and 42.6% (95% CI [41.9–43.3]) of *S. aureus* isolates across Japan and in Hokkaido, respectively, were MRSA, and the annual decline in the MRSA rate across Japan was 0.94% (95% CI [0.92–0.96]; $p < 0.01$). For JR Sapporo Hospital, 61.5% (95% CI [52.6–69.7]) of *S. aureus* isolates were MRSA in 2010, and this amount declined to 51.6% (95% CI [41.6–61.5]) in 2019, corresponding to a 1.43% (95% CI [0.42–2.43]) decrease per year ($p = 0.05$). Although the overall proportion of MRSA in JR Sapporo Hospital was greater than that across Japan or in Hokkaido, JR Sapporo Hospital exhibited a significant decrease in its MRSA rate over the 10-year period.

The antimicrobial susceptibility rates for MRSA in JR Sapporo Hospital and across Japan are summarized in Figs. 1 and 2, respectively. In JR Sapporo Hospital, the MRSA susceptibility rates for arbekacin and trimethoprim-sulfamethoxazole remained substantial (close to 100%), whereas the proportions of MRSA sensitive to gentamicin, minocycline, erythromycin, clindamycin, levofloxacin, and fosfomycin in 2019 were all higher than their 2010 levels of approximately less than 20% (Fig. 1). The estimated rates of increase are shown in Table 2. The highest rates of annual increase were observed for minocycline followed by fosfomycin, which had annual recovery rates 3.11% (95% CI [2.25–3.94]) and 2.85% (95% CI [1.83–3.85]), respectively. Additionally, a significant increasing susceptibility trend was observed for all drugs except for arbekacin, levofloxacin, and trimethoprim-sulfamethoxazole. The susceptibility of MRSA to six antimicrobial agents has been consistently examined across Japan (Fig. 2), and significant increases in MRSA susceptibility (all $p < 0.01$) were observed for gentamicin, minocycline, and clindamycin, with estimated annual increase rates of 2.51% (95% CI [1.45–3.55]), 4.01% (95% CI [2.98–5.01]), and 4.78% (95% CI [3.80–5.71]), respectively (Table 2).

Tables S1 and S2 show the AUD of different antimicrobial agents and the antimicrobial susceptibility of MRSA by department. The AUD of anti-MRSA drugs at JR Sapporo Hospital in 2019 was highest in the dermatology department (42.8) followed by the thoracic surgery (8.6), surgery (7.4), urology (6.3), orthopaedics (4.3), internal medicine (4.0), and cardiology (3.6) departments. A bivariate comparison between the AUD and MRSA antimicrobial susceptibility by department identified significant positive correlations for erythromycin ($p < 0.01$), clindamycin ($p = 0.002$), and levofloxacin ($p = 0.0005$) (Fig. 3). Thus, when anti-MRSA drugs were appropriately used where MRSA was detected, the antimicrobial susceptibility of MRSA was maintained at a high level.

## DISCUSSION

The present study examined the trend in the MRSA proportion among *S. aureus* isolates in inpatient services across Japan, in Hokkaido, and in JR Sapporo Hospital by comparing the

**Table 1 Percentages of methicillin-resistant *Staphylococcus aureus* (MRSA) for Japan, the Hokkaido region, and JR Sapporo Hospital from 2010–2019.**

| | | 2010 | 2011 | 2012 | 2013 | 2014 | 2015 | 2016 | 2017 | 2018 | 2019 | Annual % change (95% CI) in the % MRSA | p value |
|---|---|---|---|---|---|---|---|---|---|---|---|---|---|
| Japan | S. aureus | 175,145 | 210,382 | 221,239 | 231,909 | 246,030 | 349,743 | 372,787 | 383,006 | 391,316 | 400,094 | | |
| | MRSA | 100,845 | 114,933 | 117,209 | 118,539 | 120,702 | 169,528 | 177,768 | 182,619 | 185,709 | 192,320 | | |
| | % MRSA | 57.6 | 54.6 | 53.0 | 51.1 | 49.1 | 48.5 | 47.7 | 47.7 | 47.5 | 48.1 | 0.94 | <0.01 |
| | (95% CI) | [57.4–57.8] | [54.4–54.8] | [52.8–53.2] | [50.9–51.3] | [48.9–49.3] | [48.3–48.7] | [47.5–47.9] | [47.5–47.9] | [47.3–47.7] | [47.9–48.3] | [0.92–0.96] | |
| Hokkaido | S. aureus | – | – | – | – | – | 16,793 | 17,848 | 19,179 | 19,724 | 21,547 | | |
| | MRSA | – | – | – | – | – | 7,249 | 7,677 | 8,296 | 8,186 | 9,178 | | |
| | % MRSA | – | – | – | – | – | 43.2 | 43 | 43.3 | 41.5 | 42.6 | 0.26 | 0.11 |
| | (95% CI) | | | | | | [42.5–44.0] | [42.3–43.7] | [42.6–44.0] | [40.8–42.2] | [41.9–43.3] | [0.04–0.48] | |
| JR Sapporo Hospital | S. aureus | 122 | 122 | 114 | 91 | 111 | 97 | 86 | 86 | 122 | 93 | | |
| | MRSA | 75 | 85 | 75 | 62 | 56 | 57 | 45 | 49 | 72 | 48 | | |
| | % MRSA | 61.5 | 69.7 | 65.8 | 68.1 | 50.5 | 58.8 | 52.3 | 57.0 | 59.0 | 51.6 | 1.43 | 0.05 |
| | (95% CI) | [52.6–69.7] | [61.0–77.2] | [56.7–73.9] | [58.0–76.8] | [41.3–59.6] | [48.9–68.1] | [41.9–62.5] | [4.5–66.9] | [50.1–67.3] | [41.6–61.5] | [0.42–2.43] | |

**Note:**
The total numbers of S. *aureus* and MRSA isolates and the percentage of S. *aureus* isolates due to MRSA from 2010–2019. The annual percent change (the yearly rate of decline) in the percentage of S. *aureus* isolates due to MRSA; the *p*-value indicates whether the time-dependent change is statistically significant. All datasets were from inpatient records.

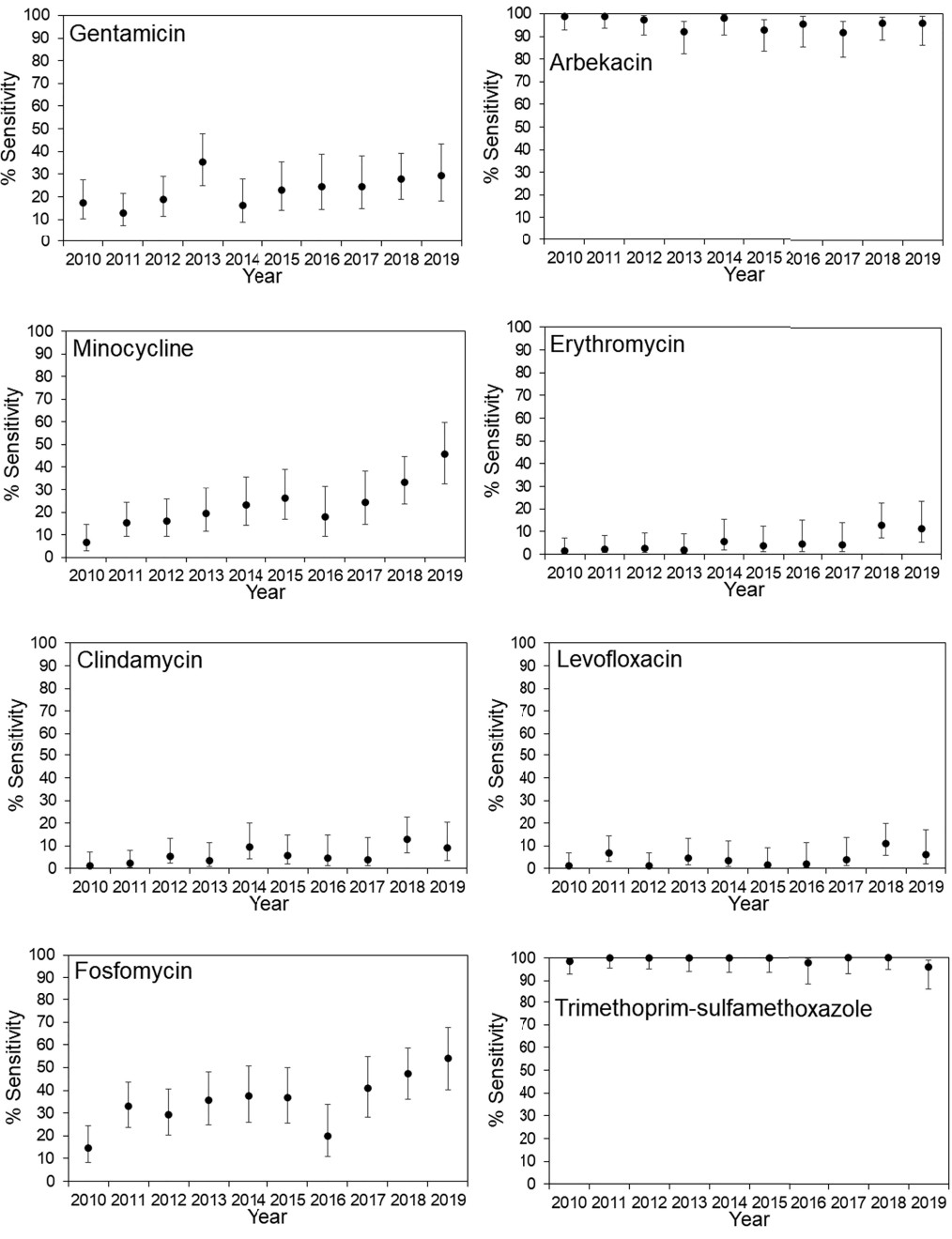

**Figure 1 Time-dependent changes in the antimicrobial susceptibility of methicillin-resistant *Staphylococcus aureus* (MRSA) in JR Sapporo Hospital from 2010–2019.** Percentage of MRSA isolates susceptible to various antimicrobial agents in JR Sapporo Hospital from 2010–2019. The yearly estimate (dot) shows the percentage of susceptible cases. The whiskers cover the 95% confidence intervals, which were calculated using Agresti's score confidence interval. The MRSA susceptibility rates to vancomycin, teicoplanin, and linezolid were all 100% for the period from 2010–2019.

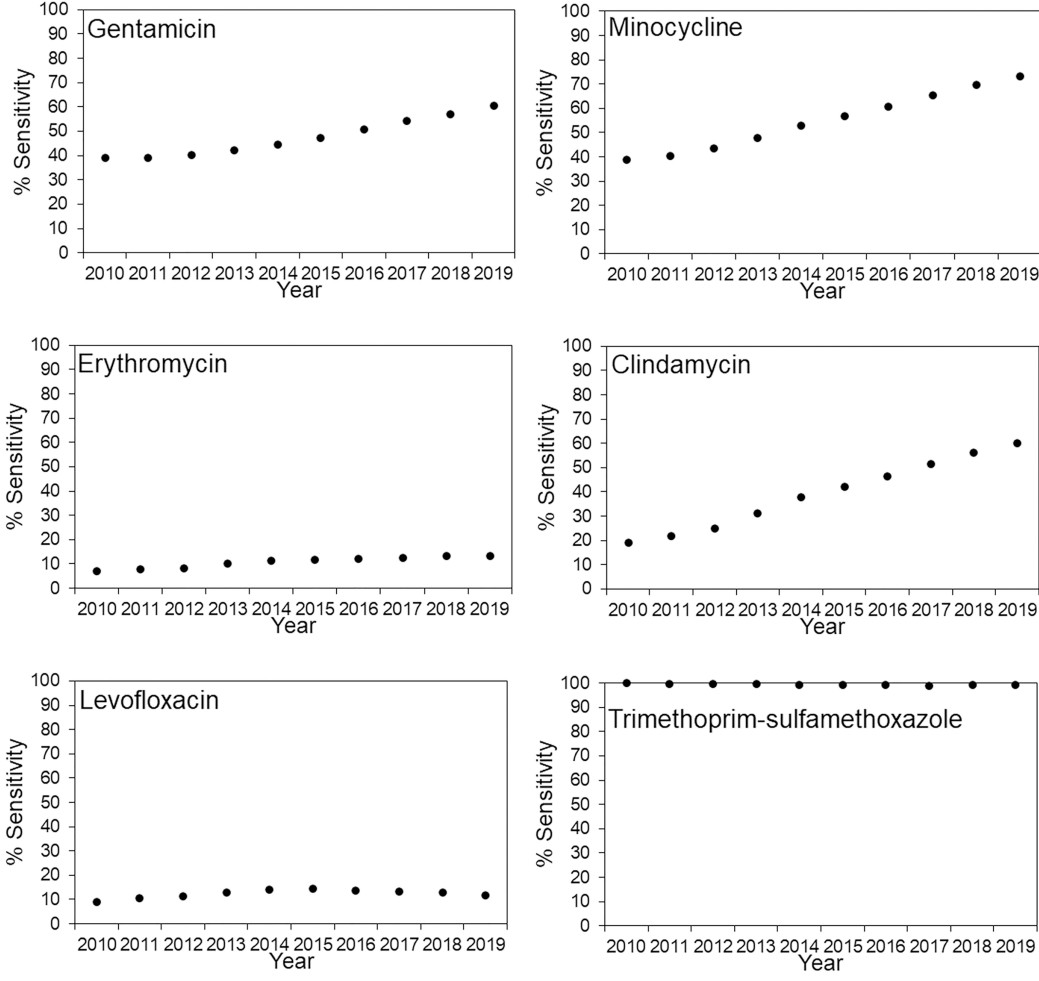

**Figure 2 Time-dependent changes in the antimicrobial susceptibility of methicillin-resistant *Staphylococcus aureus* (MRSA) across Japan from 2010–2019.** Percentage of MRSA isolates susceptible to various antimicrobial agents in JR Sapporo Hospital from 2010–2019. The yearly estimate (dot) shows the percentage of susceptible cases. The whiskers cover the 95% confidence intervals, which were calculated using Agresti's score confidence interval, but they are mostly invisible because of the large sample size of the Japan nosocomial infections surveillance (JANIS) dataset. The susceptibility rates of MRSA to arbekacin and fosfomycin could not be calculated because JANIS did not include these drugs as part of their routine monitoring. The MRSA susceptibility rates to vancomycin, teicoplanin, and linezolid are not shown because the MRSA isolates in JR Sapporo Hospital, which is the comparison group, had 100% susceptibility to these drugs.

annual rate of change in this proportion, and it also investigated the AUD and its relationship with the MRSA antimicrobial susceptibility within JR Sapporo Hospital. Overall, the data exhibit a clear decreasing trend in the proportion of MRSA among *S. aureus* isolates in Japan. Although the rate of change in MRSA in Hokkaido was not significant, a marked decreased in the MRSA proportion was observed in JR Sapporo Hospital over the 10-year study period. During this time, there was not only an overall decline in the proportion of MRSA, but also a recovery in the antimicrobial susceptibility of MRSA. Across Japan, there was an improvement in the MRSA susceptibility to gentamycin, minocycline, and clindamycin, and JR Sapporo Hospital enjoyed a significant

**Table 2 Annual percent increases in the proportion of methicillin-resistant *Staphylococcus aureus* (MRSA) susceptible to various drugs in JR Sapporo Hospital and across Japan from 2010–2019.**

| | MRSA (JR Sapporo Hospital) | | MRSA (Japan) | |
|---|---|---|---|---|
| | Annual % change (95% Cl) in cases | *p* value | Annual % change (95% Cl) in cases | *p* value |
| Gentamicin | 1.32 [0.41–2.22] | 0.045 | 2.51 [1.45–3.55] | <0.01 |
| Arbekacin | −0.54 [−0.98 to −0.05] | 0.13 | NA | |
| Minocycline | 3.11 [2.25–3.94] | <0.01 | 4.01 [2.98–5.01] | <0.01 |
| Erythromycin | 0.90 [0.50–1.28] | <0.01 | 0.79 [0.11–1.44] | 0.11 |
| Clindamycin | 0.86 [0.36–1.34] | 0.02 | 4.78 [3.80–5.71] | <0.01 |
| Levofloxacin | 0.41 [0–0.82] | 0.17 | 0.37 [0.37–1.10] | 0.49 |
| Fosfomycin | 2.85 [1.83–3.85] | <0.01 | NA | |
| Trimethoprim-sulfamethoxazole | −0.10 [−0.25 to 0.03] | <0.01 | 0.01 [−0.30 to 0.10] | 0.50 |

**Note:**
Annual percent change (the yearly rate of decline); the *p*-value indicates the statistical significance of the time-dependent change. A likelihood ratio test was implemented, comparing a null model in which there is no time-dependent trend and an alternative model in which a linear trend was incorporated. NA, not available.

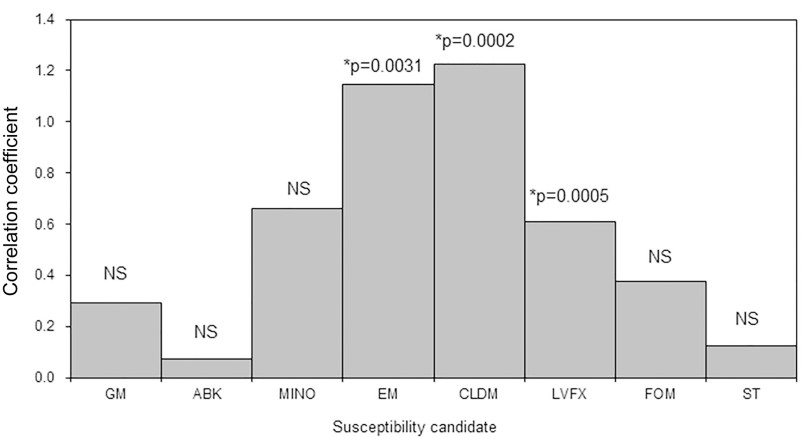

**Figure 3 Bivariate correlation analysis between the AUD and the antimicrobial susceptibility rate of methicillin-resistant *Staphylococcus aureus* (MRSA) by ward in JR Sapporo Hospital, 2019.** The correlation between the AUD and MRSA antimicrobial susceptibility rate across different inpatient wards was explored. Where MRSA was present, it is likely that anti-MRSA drugs were used. The antimicrobial susceptibilities of MRSA to erythromycin, clindamycin, and levofloxacin were significantly correlated with AUD. GM: Gentamicin, ABK: Arbekacin, MINO: Minocycline, EM: Erythromycin, CLDM: Clindamycin, LVFX: Levofloxacin, FOM: Fosfomycin, ST: Trimethoprim-sulfamethoxazole, NS: not significant.

recovery of MRSA susceptibility to gentamicin, minocycline, erythromycin, clindamycin, and fosfomycin. Exploring the relevance of these findings to antimicrobial agent use revealed that greater susceptibility to erythromycin, clindamycin, and levofloxacin was found in departments where anti-MRSA drugs were more frequently used.

An important finding of the present study is that our antibiogram dataset exhibited not only a decreasing proportion of MRSA over the study period, but also a recovery of MRSA antimicrobial susceptibility, which might have been associated with the launches of ICT in 2014 and the antimicrobial stewardship team (AST) activities in 2017. ICT and AST

activities imposed a notification system for the use of anti-MRSA drugs; additionally, broad-spectrum antibacterial agents were gradually avoided and appropriate anti-MRSA drugs were used when necessary. The observed recovery in MRSA susceptibility was seen for antimicrobial agents that are not generally the first choice for treating MRSA; however, the overall improvements observed following implementation of an antibiogram for MRSA are encouraging, particularly for cases of refractory infection and complications resulting from mixed infection, because the increased number of drug options enable the possibility of combination therapy and second-choice antimicrobial agents (especially, when the gold standard treatment using anti-MRSA drugs is not available) to be considered (*Rodvold & McConeghy, 2014*).

Another notable finding is the identification of a positive correlation between the AUD for anti-MRSA drugs and MRSA antimicrobial susceptibility, which highlights the critical importance of maintaining antimicrobial stewardship. JR Sapporo Hospital has promoted the appropriate use of antimicrobial agents since the founding of ICT, even in the absence of an infectious disease specialist. In 2017, the AST introduced a new notification system for broad-spectrum antimicrobial agents and anti-MRSA agents. Despite such a useful system, it is generally a difficult task for clinical laboratory technologists to intervene in physician decisions regarding the choice of antimicrobial agents (*Roque et al., 2014*), and physicians retain the right to make a final decision about prescriptions. Thus, to encourage antimicrobial stewardship in the hospital, infection control nurses, clinical laboratory technologists, and pharmacists have strategically built their own professional training plans, obtaining expert accreditation associated with the stewardship program, and made a special effort to find case-by-case solutions in multiple clinical departments. Presently, such effort has been set as an essential prerequisite for joining the AST in our hospital. Notably, whenever microbiological testing results were submitted to physicians, clinical laboratory technologists added a short comment, which sometimes even contained explicit clinical interpretations of the testing result. Furthermore, a proposed choice of antimicrobial agents for treatment was also presented, following a preliminary discussion with the infection disease chemotherapy pharmacist and ward-based pharmacist.

Decreases in *S. aureus* antimicrobial resistance rates have been reported in many regions worldwide, including the USA, Latin America, Canada, and Europe, although the observed rate of resistance varies widely by country (*Chang et al., 2015*; *Acree, Morgan & David, 2017*; *Arias et al., 2017*; *Kistler, Thoder & Ilyas, 2019*; *Nichol et al., 2019*; *Walter et al., 2017*). Like these other countries, Japan has shown a decreasing trend in the rate of MRSA. However, although the AMR Action Plan in Japan aimed to achieve a decrease in the antimicrobial resistance rate of *S. aureus* from 48.5% in 2015 to less than 20% by 2020, the observed 2019 rate was 48.1% across Japan, 42.6% in Hokkaido, and 51.6% in JR Sapporo Hospital, values all well above the target. An important learning point from our analysis is that the goal of antimicrobial resistance rate should be determined using a trend analysis as shown in this study; additionally, the possibility of combination therapy can also be considered to further improve antimicrobial susceptibility and lower the economic cost (*Rodvold & McConeghy, 2014*; *Davis, Van Hal & Tong, 2015*; *Tong et al., 2016*).

Four technical limitations of this work must be noted. First, the present study applied a descriptive analysis, with some statistical modelling support; thus, the causal effect of each single treatment on antimicrobial susceptibility has not been fully demonstrated. A controlled study using individual datasets should be conducted to address this gap. Second, we analysed AUD only by clinical department, because that stratification was the only available information; however, a similar analysis could be more clinically relevant if the data were classified by diagnosed disease or clinical specimen type. Third, our analysis of the time trend was not able to incorporate age, sex, or treatment history because this information was not integrated with the existing dataset. Consequently, our analysis focused purely on the temporal component alone. Fourth, because of widespread policies regarding appropriate antimicrobial agent use implemented since 2016, bacterial culture testing was recommended, which may have decreased empiric therapy overall. Thus, the proportion of MSSA could have slightly increased from around 2016.

While antimicrobial susceptibility testing at an individual level is critical in determining the appropriate choice of antimicrobial agent(s) for successful treatment, it is also vital to analyse the corresponding epidemiological data, so that the temporal dynamics of AMR can be explicitly analysed and evaluated as part of risk assessment practice. The present study identified a recovery of antimicrobial susceptibility to some antimicrobial drugs in MRSA in Japan and supports the potential for appropriate antimicrobial agent use in reviving antimicrobial susceptibility in MRSA.

## ACKNOWLEDGEMENTS

We thank Katie Oakley, PhD, from Edanz Group for editing a draft of this manuscript.

### Funding

Hiroshi Nishiura received funding from Health and Labor Sciences Research Grants (19HA1003, 20CA2024, and 20HA2007), the Japan Agency for Medical Research and Development (JP19fk0108104, JP20fk0108140 and JP20fk0108535s0101), the Japan Society for the Promotion of Science KAKENHI (17H04701 and 21H03198), the Inamori Foundation, the Japan Science and Technology Agency CREST program (JPMJCR1413); and the SICORP (e-ASIA) program (JPMJSC20U3). This study was also supported by German Federal Ministry of Health (BMG) COVID-19 Research and Development funding to the World Health Organization. The funders had no role in study design, data collection and analysis, decision to publish, or preparation of the manuscript.

### Grant Disclosures

The following grant information was disclosed by the authors:
Health and Labor Sciences Research Grants: 19HA1003, 20CA2024, and 20HA2007.
Japan Agency for Medical Research and Development: JP19fk0108104, JP20fk0108140 and JP20fk0108535s0101.
Japan Society for the Promotion of Science KAKENHI: 17H04701 and 21H03198.

Inamori Foundation.
Japan Science and Technology Agency CREST program: JPMJCR1413.
SICORP (e-ASIA) program: JPMJSC20U3.
German Federal Ministry of Health (BMG).

## Competing Interests

Hiroshi Nishiura is an Academic Editor for PeerJ.

## Author Contributions

- Yuji Koike performed the experiments, analyzed the data, prepared figures and/or tables, authored or reviewed drafts of the paper, and approved the final draft.
- Hiroshi Nishiura conceived and designed the experiments, performed the experiments, analyzed the data, prepared figures and/or tables, authored or reviewed drafts of the paper, and approved the final draft.

## Ethics

The following information was supplied relating to ethical approvals (i.e., approving body and any reference numbers):

This study was approved by the institutional review board of the Hokkaido University Graduate School of Medicine (Med 20-021).

## Data Availability

The dataset from the JANIS database can be retrieved online (*Ministry of Health, Labour and Welfare, 2020a*; https://janis.mhlw.go.jp/report/kensa_prefectures.html).

The datasets containing the antimicrobial use density (AUD) by the department and antimicrobial susceptibility in methicillin-resistant *Staphylococcus aureus* (MRSA) by the department in JR Sapporo Hospital, 2019 are available as Supplementary Tables.

## Supplemental Information

Supplemental information for this article can be found online at http://dx.doi.org/10.7717/peerj.11644#supplemental-information.

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
