# Peer review of "Recovery of antimicrobial susceptibility in methicillin-resistant Staphylococcus aureus (MRSA): a retrospective, epidemiological analysis in a secondary care hospital, Sapporo, Japan"

_PeerJ, doi:10.7717/peerj.11644_

## Round 0.1 · original submission · Minor Revisions

I am looking forward to receiving your revised manuscript.

Kind regards
Elisabeth Grohmann

·

Basic reporting

No comment

Experimental design

No comment

Validity of the findings

No comment

Additional comments

1. Please introduce the abbreviations on their first use and only once in the manuscript, for example, the abrreviation 'AUD'. Make sure the abbreviations in the abstract are defined as well.
2. Make sure bacterial genus and species names are italicized.
3. Please check the grammar of the manuscript for minor errors.

Reviewer 2 ·

Basic reporting

No comment

Experimental design

No comment

Validity of the findings

No comment

Additional comments

The manuscript entitled „Recovery of antimicrobial susceptibility in methicillin-resistant Staphylococcus aureus (MRSA): Epidemiological analysis using antibiogram in a secondary care hospital, Sapporo, Japan (#55931) deals with the significant problem of protecting antibiotics through their rational use. This manuscript can be published in Peer J but major revision is necessary.

In manuscript instead "MRSA cases" should be "MRSA isolates",
Line 29: please add "AUD" abbreviation,
Line 98: “The present study aimed to clarify the percentage of S. aureus cases caused by MRSA” - this is not clear,
Line 108: “The hospital is within a 10-minute walking distance from the main station, Sapporo station” - whether this information is relevant and necessary,
The title of Table 2 is not appropriate because it does not correspond to the information contained in this table. In this table showed significant increases of MRSA susceptibility for some antibiotics in tested range of time.
Figure 1 and 2 – instead “% sensitivity” should be “% of sensitive MRSA isolates”.

Reviewer 3 ·

Basic reporting

I was not able to access the JANIS dataset from the URL provided (https://janis.mhlw.go.jp/report/prefectures.asp): it just gave an error. Please check that a functional URL is provided.

Experimental design

Some description of antimicrobial susceptibility testing methods used should be provided, at least those for primary institution (JR Sapporo Hospital). What were the susceptibility breakpoint values used for each antimicrobial agent, and were consistent breakpoints used across Japan during the study period?

Validity of the findings

Line 175-176: the data for Hokkaido is incomplete and the change is not significant, so should be removed from this sentence.

Line 217: There was NOT a significant recovery of susceptibility to trimethoprim-sulfamethoxazole at JRSH; the data indicates a significant DECREASE in susceptibility (Table 2). Moreover, based on the Fig. 1 graph, I cannot see how this change can be statistically significant; please check this.

Additional comments

The manuscript is generally clearly written and logically organised, with appropriate reference to relevant literature. The figures and tables summarise the data well. The study satisfactorily compares changes in MRSA susceptibility at JR Sapporo Hospital over a decade with those observed across Japanese hospitals more broadly. As outline below, there are a few areas where it could be improved.

Methods are not usually stated in a title; I’d suggest removing “using antibiogram” from the title.

Line 70: I think “social” should be replaced by “significant health” or something similar.

Is “MRSA percentage” (line 25 and elsewhere) widely understood terminology? It’s not completely obvious percentage of what? It might be worth explaining that it is the percentage of all S. aureus isolates that are resistant to methicillin. Similarly, “the percentage of MRSA that were antimicrobial resistant” (line 147) is also lazy, since all MRSA are antimicrobial resistant (to methicilin). What is actually meant is “the percentage of MRSA that were also resistant to other antimicrobial agents”.

Lines 84-87 and 91-92: I find references to the author’s employment unnecessary, and raise the spectre that interpretations might be influenced by the desire to justify their position.

Lines 105-108: Much of this information is unnecessary.

Line 257: It is not clear if all S. aureus or MRSA is meant. If based on the data presented here it should be MRSA, if the former then suitable references should be cited.

Line 279: remove “be”.


There are a few places where conclusions should be presented more conservatively in my view…

Lines 37 and 284: “revealed the critical potential” might more appropriately be “support the potential”.

Line 223: “is likely to” could more appropriately be “might ”, since there is no direct evidence of a link.

Line 234: “critical” is too strong for the data presented; “interesting” or even “notable” would be more appropriate.

---

## Round 0.2 · accepted · Accept

Dear Dr. Nishiura,

Your manuscript has considerably improved via the revision. It is now ready for being published in the journal.

Congratulations!

Kind regards,
Elisabeth Grohmann